# REAL-FAKE: EFFECTIVE TRAINING DATA SYNTHESIS THROUGH DISTRIBUTION MATCHING

**Jianhao Yuan**[†◇]  **Jie Zhang**[‡]  **Shuyang Sun**[†]  **Philip Torr**[†]  **Bo Zhao**[◇]
[†]University of Oxford  [‡] ETH Zurich  [◇] Beijing Academy of Artificial Intelligence

## ABSTRACT

Synthetic training data has gained prominence in numerous learning tasks and scenarios, offering advantages such as dataset augmentation, generalization evaluation, and privacy preservation. Despite these benefits, the efficiency of synthetic data generated by current methodologies remains inferior when training advanced deep models exclusively, limiting its practical utility. To address this challenge, we analyze the principles underlying training data synthesis for supervised learning and elucidate a principled theoretical framework from the distribution-matching perspective that explicates the mechanisms governing synthesis efficacy. Through extensive experiments, we demonstrate the effectiveness of our synthetic data across diverse image classification tasks, both as a replacement for and augmentation to real datasets, while also benefits such as out-of-distribution generalization, privacy preservation, and scalability. Specifically, we achieve 70.9% top1 classification accuracy on ImageNet1K when training solely with synthetic data equivalent to $1 \times$ the original real data size, which increases to 76.0% when scaling up to 10 $\times$ synthetic data[1].

## 1 INTRODUCTION

Large-scale annotated datasets (Deng et al., 2009; Lin et al., 2014) are essential in deep learning for image recognition, providing the comprehensive information that models need to effectively discover patterns, learn representations, and generate accurate predictions. However, manually collecting such datasets is time-consuming and labor-intensive, and may cause privacy concerns. Given these challenges, training data synthesis offers a promising alternative to traditional data collection for augmenting or replacing original real datasets.

Among numerous image synthesis strategies, deep generative models have gained significant attention, primarily due to their capacity to produce high-fidelity images. Early studies (Besnier et al., 2020; Li et al., 2022; Zhang et al., 2021; Zhao & Bilen, 2022) utilize Generative Adversarial Networks (GANs) to synthesize annotated training data for image classification and segmentation. Recently, more works have focused on synthesizing training data with the powerful diffusion models for self-supervised pre-training (Tian et al., 2023), transfer learning (He et al., 2022), domain generalization (Yuan et al., 2022; Bansal & Grover, 2023), and supervised image classification (Azizi et al., 2023; Sarıyıldız et al., 2023; Lei et al., 2023). However, despite the extensive research, a notable performance gap persists when comparing the performances of models trained on synthetic data to those trained on real data. A primary reason for this discrepancy is the misalignment between the synthetic and real data distributions, even the diffusion model is trained on web-scale datasets, as illustrated in Fig. 1(a). While previous works have attempted to address this issue through heuristic-driven approaches such as prompt engineering (Sarıyıldız et al., 2023; Lei et al., 2023) and the expensive inversion approaches (Zhao & Bilen, 2022; Zhou et al., 2023), these solutions are neither sufficient nor efficient. Furthermore, there is an absence of theoretical frameworks that can adequately explain and analyze the efficacy of synthetic training data in a principled way.

To further enhance the quality and utility of synthetic training data produced by deep generative models, we present a theoretical framework for training data synthesis from a distribution-matching

---

[1]Code released at: `https://github.com/BAAI-DCAI/Training-Data-Synthesis`. Corresponding to Bo Zhao ⟨bozhaonanjing@gmail.com⟩

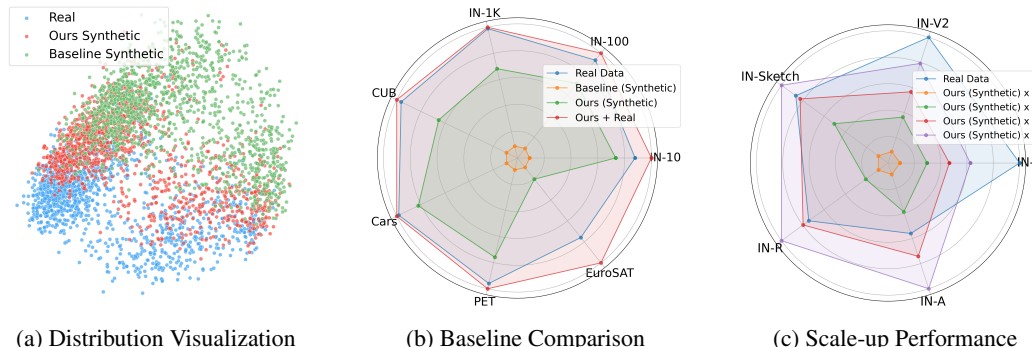

(a) Distribution Visualization   (b) Baseline Comparison   (c) Scale-up Performance

Figure 1: **Left:** Visualization of the synthetic and real ImageNet data distribution using the first two principal components of features extracted by the CLIP image encoder. Our synthetic data better aligns with real data distribution than the baseline (vanilla Stable Diffusion). **Middle:** Our synthetic data achieves better performance compared with the baseline, and can effectively augment real data across all datasets. **Right:** Scaling up synthetic training data can improve the image classification performances in both in-distribution and out-of-distribution (OOD) tasks, even outperform training with real data in OOD tasks.

perspective. Specifically, starting from the first principle of supervised learning (Sutskever, 2023; Cunningham et al., 2008), we recast the training data synthesis as a distribution matching problem. This emphasizes two primary principles of synthetic data: **(1) The distribution discrepancy between target and synthetic data**, and **(2) The cardinality of the training set**.

Building on this foundation, we implement the framework on the state-of-the-art text-to-image diffusion model, Stable Diffusion (Rombach et al., 2022), to undertake a careful analysis and refinement of the training objectives, condition generation, and prior initialization to achieve a better alignment between synthetic and target data distributions. We empirically validate our theoretical framework and synthesis method across diverse benchmarks, covering various scenarios: (1) training exclusively with synthetic data, (2) augmenting the real training data, and (3) evaluating the scaling law between synthetic data and performance. In particular, for ImageNet1k classification using ResNet50 (He et al., 2016), training solely with synthetic data equivalent to $1 \times$ the original real data size yielded a 70.9% Top1 classification accuracy, which increases to 76.0% when using $10 \times$ synthetic data. Additionally, we explore (1) out-of-distribution (OOD) generalization and (2) privacy preservation when learning with synthetic data, and present promising results. Beyond advancing the state of the art, our findings offer insights into potential strategies for refining the training data synthesis pipeline. Our primary contributions are as follows:

- Introducing a principled distribution matching framework for training data synthesis, emphasizing two foundational aspects that drive the effectiveness of synthetic data.
- Employing the state-of-the-art text-to-image diffusion model, with a comprehensive analysis and refinement of its components, to design an effective training data synthesis strategy.
- Advancing the state-of-the-art in training data synthesis for image classification tasks, while demonstrating the advantages in OOD generalization and privacy preservation.

## 2 BACKGROUND

### 2.1 TRAINING DATA SYNTHESIS

Synthesizing informative training samples remains a compelling yet challenging area of research. One line of work focuses on synthesis through pre-trained deep generative models. Early attempts (Zhang et al., 2021; Li et al., 2022; Zhao & Bilen, 2022) explore Generative Adversarial Networks (GANs) (Goodfellow et al., 2014; Brock et al., 2018) for informative and annotated training sample synthesis. Recently, diffusion models have gained more attention. He et al. (2022); Tian et al. (2023) employ synthetic data from text-conditioned diffusion models for self-supervised pre-training and few/zero-shot learning, highlighting the transfer learning capacity of synthetic training data. Also, Yuan et al. (2022); Bansal & Grover (2023); Vendrow et al. (2023) demonstrate the utility of synthetic

data in out-of-distribution settings by augmenting and diversifying the training dataset. Additionally, Sarıyıldız et al. (2023); Lei et al. (2023); Azizi et al. (2023) utilize Stable Diffusion (Rombach et al., 2022) and Imagen (Saharia et al., 2022) with prompt engineering to synthesize class-conditioned samples, showcasing the potential of using synthetic data solely for training classification models.

Despite the promising results in various tasks, a noticeable performance gap remains between models trained on real and synthetic datasets. This difference can be attributed to the misalignment between the distributions of the synthetic training data and the target downstream data. Existing research (Sarıyıldız et al., 2023; Lei et al., 2023) employs prompt engineering to bridge the domain gap, which is insufficient. Zhou et al. (2023) implement diffusion inversion to obtain synthetic images close to real ones, which is expensive and unscalable to large datasets. To understand the mechanisms underlying the efficacy of synthetic data, we aim to find the theoretical principle and further tackle the challenges associated with suboptimal training data synthetic.

Another line of work focuses on synthesizing informative training samples by distilling the large real dataset into smaller synthetic one, i.e., dataset distillation. These methods optimize synthetic images (pixels) through the minimization of meta-loss (Wang et al., 2018), gradient matching loss (Zhao et al., 2021), trajectory matching loss (Cazenavette et al., 2022) and distribution matching loss (Zhao & Bilen, 2023; Zhao et al., 2023). In particular, distribution matching approaches have gained prominence due to their model-agnostic nature. However, the intricate and expensive optimization processes of dataset distillation often pose challenges in scalability. Despite these challenges, the principled approaches of dataset distillation have proven effective and inspired our work.

## 2.2 Diffusion Probabilistic Models

Diffusion Probabilistic Models (Sohl-Dickstein et al., 2015; Ho et al., 2020; Nichol & Dhariwal, 2021) are latent variable models that learn a data distribution $q(\boldsymbol{x}_0)$ by reversing a gradual noising process. They define a forward diffusion Markov process $q\left(\boldsymbol{x}_{1:T} \mid \boldsymbol{x}_0\right) = \prod_{t=1}^{T} q\left(\boldsymbol{x}_t \mid \boldsymbol{x}_{t-1}\right)$ that uses a handcrafted Gaussian transition kernel, $q\left(\boldsymbol{x}_t \mid \boldsymbol{x}_{t-1}\right) = \mathcal{N}\left(\boldsymbol{x}_t; \sqrt{1 - \beta_t}\boldsymbol{x}_{t-1}, \beta_t\mathbf{I}\right)$, with a noise schedule $\beta_t \in (0, 1)$, to transform the data distribution into a known prior Gaussian distribution. Then, a reverse Markovian process $p_{\boldsymbol{\theta}}\left(\boldsymbol{x}_{0:T}\right) = p\left(\boldsymbol{x}_T\right) \prod_{t=1}^{T} p_{\boldsymbol{\theta}}\left(\boldsymbol{x}_{t-1} \mid \boldsymbol{x}_t\right)$ is learned to gradually remove noise added in the forward process, with Gaussian transitions parameterized by a neural network: $p_{\boldsymbol{\theta}}\left(\boldsymbol{x}_{t-1} \mid \boldsymbol{x}_t\right) = \mathcal{N}\left(\boldsymbol{x}_{t-1}; \boldsymbol{\mu}_{\boldsymbol{\theta}}\left(\boldsymbol{x}_t, t\right), \boldsymbol{\Sigma}_{\boldsymbol{\theta}}\left(\boldsymbol{x}_t, t\right)\right)$. The training objective of the diffusion model is to minimize the KL-divergence between the joint distributions of the forward and backward processes. This can be further simplified to optimizing the standard variational bound on the negative log-likelihood of the model (Yang et al., 2022) (detailed proof can be found in Apx. A).

$$\text{KL}\left[q\left(\boldsymbol{x}_0, \boldsymbol{x}_1, \cdots, \boldsymbol{x}_T\right) \| p_{\boldsymbol{\theta}}\left(\boldsymbol{x}_0, \boldsymbol{x}_1, \cdots, \boldsymbol{x}_T\right)\right] \geq \mathbb{E}\left[-\log p_{\boldsymbol{\theta}}\left(\boldsymbol{x}_0\right)\right] \quad (1)$$

In practice, by marginalizing over the intermediate sampling steps, an analytic form of the forward process can be obtained: $q\left(\boldsymbol{x}_t \mid \boldsymbol{x}_0\right) = \mathcal{N}\left(\boldsymbol{x}_t; \sqrt{\bar{\alpha}_t}\boldsymbol{x}_0, (1 - \bar{\alpha}_t)\mathbf{I}\right)$, where $\alpha_t := 1 - \beta_t$ and $\bar{\alpha}_t := \prod_{s=0}^{t} \alpha_s$. By further setting the variance in the reverse process to a non-learnable constant $\sigma_t$ and choosing a specific parameterization of $\epsilon_{\boldsymbol{\theta}}$ to predict the added noise $\epsilon$, the overall training objective becomes:

$$\mathbb{E}_{\boldsymbol{x}, \boldsymbol{\epsilon} \sim \mathcal{N}(0,1), t}\left[\frac{\beta_t^2}{2\sigma_t^2 \alpha_t(1 - \bar{\alpha}_t)} \left\|\boldsymbol{\epsilon} - \boldsymbol{\epsilon}_{\boldsymbol{\theta}}\left(\sqrt{\bar{\alpha}_t}\boldsymbol{x}_0 + \sqrt{1 - \bar{\alpha}_t}\boldsymbol{\epsilon}, t\right)\right\|^2\right], \quad (2)$$

where $\boldsymbol{x}_t = \sqrt{\bar{\alpha}_t}\boldsymbol{x}_0 + \sqrt{1 - \bar{\alpha}_t}\boldsymbol{\epsilon}$ is derived from the reparameterization of the marginal distribution $q\left(\boldsymbol{x}_t \mid \boldsymbol{x}_0\right)$. This objective (Ho et al., 2020) aligns with the denoising score matching loss of Score-based Generative Models (SGM) (Song & Ermon, 2019) when appropriately reweighted.

Leveraging the theoretical foundation, in this study, we focus on the state-of-the-art Latent Diffusion Model (LDM), Stable Diffusion (Rombach et al., 2022), to provide an alternative interpretation of its training objectives and generative sampling process from a distribution matching perspective for training data synthesis.

## 3 Training Data Synthesis: A Distribution Matching Perspective

The goal of training data synthesis is to generate synthetic data from the target distribution $D := q(x, y)$ of data $x$ and annotation $y$. In supervised training (Sutskever, 2023; Cunningham et al., 2008),

the difference between training and testing error over the whole target distribution $D$ is bounded by the inverse of the square root of the sampled training set cardinality $|S|$[2]. Please refer to Apx. B for the detailed proof.

$$\Pr_{S \sim D^{|S|}} \left[ \text{Test}_D(f) - \text{Train}_S(f) \leq \sqrt{\frac{\log |\mathcal{F}| + \log 1/\delta}{|S|}} \forall f \in \mathcal{F} \right] > 1 - \delta \tag{3}$$

For training data synthesis given a fixed model space $\mathcal{F}$ (i.e. fixed model structure with finite parameters), a salient takeaway from Eq. (3) is the identification of two pivotal factors: **(1)** *Training and testing data distribution discrepancy* **(2)** *Cardinality of training set*. This formalizes the first principle of training data synthesis: *With infinite training samples from target distribution, the testing error will converge to the minimized training error.*

However, direct sampling from data distribution can be intractable, we instead learn a generative model $p_{\boldsymbol{\theta}}$, parameterized with $\boldsymbol{\theta}$, that can synthesize data following the same distribution, i.e., $p_{\boldsymbol{\theta}}(\boldsymbol{x}, y) = q(\boldsymbol{x}, y)$ (Bishop, 2006). This effectively converts the problem of informative training data synthesis into a distribution matching problem. We can further reframe such distribution matching problems as $q(y|\boldsymbol{x})q(\boldsymbol{x}) = p_{\boldsymbol{\theta}}(y|\boldsymbol{x})p_{\boldsymbol{\theta}}(\boldsymbol{x})$ (Li et al., 2017). This allows us to separate the problem of matching a joint data-annotation distribution as two sub-problems: **(1)** data distribution matching $q(\boldsymbol{x}) = p_{\boldsymbol{\theta}}(\boldsymbol{x})$; **(2)** conditioned class likelihood matching $q(y|\boldsymbol{x}) = p_{\boldsymbol{\theta}}(y|\boldsymbol{x})$. Under the classification protocol, the former ensures in-distribution data synthesis, and the latter ensures a robust decision boundary between classes. Overall, the objective of training data synthesis for supervised learning can be framed as the following optimization:

$$S^* = \underset{S \sim p_{\boldsymbol{\theta}}(\boldsymbol{x}, y)}{\arg \min} \; (D(q(\boldsymbol{x}), p_{\boldsymbol{\theta}}(\boldsymbol{x})) + D(q(y|\boldsymbol{x}), p_{\boldsymbol{\theta}}(y|\boldsymbol{x})) - \lambda|S|) \tag{4}$$

where $S^*$ denotes the optimal synthetic data sampled from the learned distribution $S \sim p_{\boldsymbol{\theta}}(\boldsymbol{x}, y)$. $D(\cdot, \cdot)$ is a distance measure between two distributions. The regularization term $\lambda|S|$ with $\lambda \in \mathbb{R}^+$ encourages a larger training set. Fortunately, both distribution matching problems can be solved with deep generative models. In particular, with the diffusion model, the data distribution is learned through a denoising process, and the class likelihood can be modeled through classifier or classifier-free guidance (Dhariwal & Nichol, 2021; Ho & Salimans, 2022).

Based on this theoretical framework, we perform an analysis of each component of the diffusion model in the context of distribution matching and propose potential improvements. Specifically, we introduce a distribution matching-centric synthesis framework tailored for training data synthesis, including three aspects of (1) feature distribution alignment (2) conditioned visual guidance (3) latent prior initialization.

## 3.1 Distribution Matching with Maximum Mean Discrepancy

Firstly, we quantify the distribution discrepancy between target and synthetic data. Although the objective of KL-divergence minimization in the diffusion model has implicitly provided an upper bound for the data distribution matching (Yang et al., 2022), it is observed to be loose (Kingma et al., 2021) due to the gap between the bound and the negative log-likelihood Eq. (1). More importantly, the discrete nature of empirical distribution (i.e. data samples) in the context of training data synthesis further makes it a suboptimal measure. Instead, we measure this discrepancy with the Maximum Mean Discrepancy (MMD) (Gretton et al., 2008), which alleviates the problem of potentially biased discrete distribution as a sample test and is widely and successfully applied in dataset distillation Zhao & Bilen (2023) because of its non-parametric nature and simplicity. At the end, we find a mathematical equivalence between the training objective of diffusion model and the minimization of the MMD upper bound under certain assumptions, which allows us to further relax the variational bound and better align the data distribution in the feature space.

Consider a (real) target dataset $\mathcal{R} = \left\{ (k_1, y_1), \ldots, (k_{|\mathcal{R}|}, y_{|\mathcal{R}|}) \right\}$ and a synthetic dataset $\mathcal{S} = \left\{ (s_1, y_1), \ldots, (s_{|\mathcal{S}|}, y_{|\mathcal{S}|}) \right\}$. Our objective is to minimize the MMD between target data distribution

---

[2]The probability that the error is lower than a bound, which explicitly depends on the cardinality of the training set, is guaranteed to be higher than $1 - \delta$, where $\delta$ is small.

$q(x)$ and the synthetic data distribution $p_{\boldsymbol{\theta}}(x)$ represented by some feature extractor $\psi \in \mathcal{F}$.

$$\text{MMD}[\mathcal{F}, p, q] = \sup_{\|\psi_\vartheta\|_{\mathcal{H}} \leq 1} \left( \mathbb{E}_q \left[ \psi(\mathcal{R}) \right] - \mathbb{E}_p \left[ \psi(\mathcal{S}) \right] \right), \tag{5}$$

where $\psi$ represents a function residing within a unit ball in the universal Reproducing Kernel Hilbert Space $\mathcal{H}$ (RKHS) (Hilbert, 1904). By following the reproductive properties of RKHS and empirically estimating the expectations for all distributions, we can simplify Eq. (5) to Eq. (6):

$$\text{MMD}^2[\mathcal{F}, p, q] = \| \frac{1}{|\mathcal{T}|} \sum_{i=1}^{|\mathcal{T}|} \psi_\vartheta(k_i) - \frac{1}{|\mathcal{S}|} \sum_{j=1}^{|\mathcal{S}|} \psi_\vartheta(\boldsymbol{s}_j) \|_{\mathcal{H}}^2. \tag{6}$$

In Latent Diffusion Model, $\psi$ can be conceptualized as the Variational Autoencoder (VAE) with the latent embedding serving as the feature map for MMD computation. The distributions of target and synthetic data can be effectively approximated by the initial noise-free latent embedding $\boldsymbol{x}_0$ and the predicted denoised latent embedding $\boldsymbol{x}_{\boldsymbol{\theta}}(\boldsymbol{x}_t, t)$. Assuming that MMD is computed within the same batch, where $|\mathcal{R}| = |\mathcal{S}| = |\mathcal{N}|$, our objective can be further refined as $\| \frac{1}{|\mathcal{N}|} \sum_{i=1}^{|\mathcal{N}|} (\boldsymbol{x} - \boldsymbol{x}_{\boldsymbol{\theta}}(\boldsymbol{x}_t, t)) \|_{\mathcal{H}}^2$. Following Ho et al. (2020), choosing the parameterization with $\epsilon_{\boldsymbol{\theta}}$ as a predictor of added noise $\epsilon$ from $x_t$ in denoising, allows us to frame our distribution matching objective (MMD) as Eq. (7):

$$L_{MMD} := \| \frac{1}{|\mathcal{N}|} \sum_{i=1}^{|\mathcal{N}|} (\epsilon - \epsilon_{\boldsymbol{\theta}}(\boldsymbol{x}_t, t)) \|_{\mathcal{H}}^2 \leq \frac{1}{|\mathcal{N}|} \sum_{i=1}^{|\mathcal{N}|} \| (\epsilon - \epsilon_{\boldsymbol{\theta}}(\boldsymbol{x}_t, t)) \|_{\mathcal{H}}^2 = L_{Diffusion}. \tag{7}$$

Since the mean and norm operations are both convex functions on the left-hand side, by applying Jensen's inequality, we obtain an upper bound on the original distribution matching objective under equal weighting to each time step. Note that, this upper bound expression resembles the diffusion model training loss Eq. (2). This indicates when training the diffusion model, we implicitly optimize an upper bound of the MMD between synthetic and real data distribution. Moreover, this allows us to directly optimize the objective in Eq. (7) to mitigate the losseness in the variational bound. We use this objective to augment the original diffusion model loss during finetuning, which could ensure a more aligned feature distribution under the MMD measure.

## 3.2 Conditioned Generation via Text-Vision Guidance

Beyond the general feature-level distribution matching, i.e., $q(\boldsymbol{x}) = p_{\boldsymbol{\theta}}(\boldsymbol{x})$, a pivotal aspect of training data synthesis is to ensure a congruent conditional class distribution with well-defined decision boundaries, i.e., $q(y|\boldsymbol{x}) = p_{\boldsymbol{\theta}}(y|\boldsymbol{x})$. Classifier(-free) guidance (Dhariwal & Nichol, 2021; Ho & Salimans, 2022) plays a crucial role in the conditioned sampling process in the diffusion model. Under SGM framework, to match the conditioned class likelihood, we can equivalently match the score function of each distribution, i.e. $\nabla_{\boldsymbol{x}_t} \log q(y|\boldsymbol{x}) = \nabla_{\boldsymbol{x}_t} \log p_{\boldsymbol{\theta}}(y|\boldsymbol{x})$, which is estimated through the noise prediction $\epsilon_{\boldsymbol{\theta}}(\boldsymbol{x}_t)$ by reformulating the conditional score function as Eq. (8):

$$\nabla_{\boldsymbol{x}_t} \log p_{\boldsymbol{\theta}}(y \mid \boldsymbol{x}_t) = \nabla_{\boldsymbol{x}_t} \log p_{\boldsymbol{\theta}}(\boldsymbol{x}_t \mid y) - \nabla_{\boldsymbol{x}_t} \log p_{\boldsymbol{\theta}}(\boldsymbol{x}_t) = \frac{1}{\sqrt{1 - \bar{\alpha}_t}} (\epsilon_{\boldsymbol{\theta}}(\boldsymbol{x}_t, y) - \epsilon_{\boldsymbol{\theta}}(\boldsymbol{x}_t)) \tag{8}$$

Most works of training data synthesis align the conditioned distribution via text-prompt engineering with class-level description (He et al., 2022), instance-level description (Lei et al., 2023), and lexical definition (Sarıyıldız et al., 2023). Following (Lei et al., 2023), we incorporate the class name with the BLIP2 (Li et al., 2023) caption of each instance as the text prompt. Moreover, while text condition offers certain adaptability, it ignores intrinsic visual information, including both low-level ones, e.g., exposure and saturation, and high-level ones, e.g. co-occurrence of objects and scenes. To address this, we adopt a more direct prompting strategy by conditioning on image features. In particular, we extract image features encoded with the CLIP (Radford et al., 2021) image encoder, compute the mean of image embeddings for randomly sampled images of a class, and estimate the intra-class feature distribution, i.e. the mean feature. This is then concatenated with the text embeddings for jointly finetuning the diffusion model using LoRA (Hu et al., 2021), thus injecting the extra conditional control into the cross-attention layer of the denoising UNet (Ronneberger et al., 2015). The resultant multi-modal condition (embedding) takes form in `"photo of [classname], [Image Caption], [Intra-class Visual Guidance]"`.

### 3.3 LATENT PRIOR INITIALIZATION

Another key factor ensuring the data distribution matching of synthetic data is the latent prior $p_{\boldsymbol{\theta}}(\boldsymbol{x}_T)$ in the sampling process of LDM. It acts as the informative guide for the reverse diffusion process, governed by Langevin dynamics (Song & Ermon, 2019; Parisi, 1981), as shown in Eq. (9):

$$\boldsymbol{x}_{t-1} = \boldsymbol{x}_t + \sqrt{2\alpha_t}\mathbf{z}_t - \alpha_t \nabla_{\boldsymbol{x}_t} \log p_{\boldsymbol{\theta}}(\boldsymbol{x}_t|\boldsymbol{x}_T), \mathbf{z}_t \sim \mathcal{N}(\mathbf{0}, \mathbf{I}), \tag{9}$$

where $p_{\boldsymbol{\theta}}(\boldsymbol{x}_t|\boldsymbol{x}_T)$ represents the conditional distribution of $\boldsymbol{x}_t$ given the latent prior $\boldsymbol{x}_T$, and the corresponding score function $\nabla_{\boldsymbol{x}_t} \log p_{\boldsymbol{\theta}}(\boldsymbol{x}_t|\boldsymbol{x}_T)$ guides the Langevin dynamics towards regions of higher probability in the data distribution, ensuring the synthetic samples align closely with the target distribution. While diffusion models often employ a Gaussian distribution as the initial prior $\boldsymbol{x}_T \sim q(\boldsymbol{x}_T) := \mathcal{N}(\boldsymbol{x}_T; \mathbf{0}, \mathbf{I})$, recent studies using latent inversion (Zhou et al., 2023; Lian et al., 2023) and learning-based prior rendering (Liao et al., 2023) have highlighted the advantages of informative non-Gaussian latent priors with better sampling speed and synthesis quality. However, obtaining such an informative prior can be expensive due to intensive computation or need for external architecture. Instead, following (Meng et al., 2021), we leverage the VAE encoder to obtain the latent code of specific real samples, which is extremely cheap. This also provides an informative latent prior initialization closely aligned with the target distribution, resulting in better synthetic samples.

## 4 EXPERIMENTS

**Settings.** We empirically evaluate the proposed distribution-matching framework and assess the utility of synthetic training data. We explore the application of synthetic data in various supervised image classification scenarios: **(1)** Replacing the real training set (Sec. 4.1), **(2)** Augmenting the real training set (Sec. 4.2), and **(3)** Evaluating the scaling law of synthetic training data (Sec. 4.3). We aim to validate two main factors identified in Eq. (4): *better alignment between target and synthetic data* (in scenarios 1 and 2) and the advantages of *larger training set cardinality* (in scenario 3). Then, we further explore the **(4)** Out-of-distribution generalization (Sec. 4.4) and **(5)** Privacy-preservation (Sec. 4.5) of synthetic training data. For all experiments, we finetune Stable Diffusion 1.5 (SDv1.5) (Rombach et al., 2022) with LoRA.

**Datasets.** We conduct benchmark experiments with ResNet50 (He et al., 2016) across three ImageNet datasets: ImageNette (IN-10) (Howard, 2019), ImageNet100 (IN-100) (Tian et al., 2020), and ImageNet1K (IN-1K) (Deng et al., 2009). Beyond these, we also experiment with several fine-grained image classification datasets, CUB (Wah et al., 2011), Cars (Krause et al., 2013), PET (Parkhi et al., 2012), and satellite images, EuroSAT (Helber et al., 2018).

More details of dataset specifications (Apx. D), data synthesis (Apx. E), model training (Apx. F), and experiment setting (Apx. G) are provided in the Appendix.

### 4.1 IMAGE CLASSIFICATION WITH SYNTHETIC DATA ONLY

We begin by assessing the informativeness of synthetic training data as a replacement for real training data on image classification tasks. To replace the real training set, we synthesize training samples with the same number as the real training set in each dataset.

As shown in Tab. 1, in comparison to real data, our synthetic data reduces the performance gap to less than 3% for small-scale dataset IN-10. In the context of the more challenging large-scale dataset, IN-1K, the performance differential remains under 10%. Regarding training data synthesis methods, our method outperforms all state-of-the-art techniques across all benchmarks. It is crucial to highlight that our synthetic data exhibits improvements of 16.8% and 28.0% in IN-1K top-1 accuracy, compared to CiP (Lei et al., 2023) and FakeIt (Sarıyıldız et al., 2023) respectively, given the same generative model backbone. In fine-grained classification tasks, our method demonstrates a more significant improvement compared to the state-of-the-art method (FakeIt), which can be attributed to the need for a more aligned decision boundary.

**Ablation Study.** We next perform a more comprehensive ablation study to evaluate the efficacy of our proposed enhancements: distribution matching objective (Sec. 3.1), conditioned visual guidance (Sec. 3.2), and latent prior initialization (Sec. 3.3). Due to computational cost, we conducted the

Table 1: **Synthetic Image Classification Performance** Top-1 accuracies of ResNet50 reported on seven datasets. The upper part indicates training with synthetic data only, and the lower part indicates joint training with combined real and synthetic data.

| | Model | IN-10 | IN-100 | IN-1K | CUB | Cars | PET | EuroSAT |
|---|---|---|---|---|---|---|---|---|
| *Training **without** real data* | | | | | | | | |
| BigGAN (Brock et al., 2018) | BigGAN | - | - | 42.7 | | | | |
| VQ-VAE-2 (Razavi et al., 2019) | VQ-VAE | - | - | 54.8 | | | | |
| CDM (Ho et al., 2022) | CDM | - | - | 63.0 | | | | |
| FakeIt (Sarıyıldız et al., 2023) | SDv1.5 | - | - | 42.9 | 33.7 | 47.1 | 75.9 | 94.0 |
| Imagen (Azizi et al., 2023) | Imagen | - | - | 69.2 | | | | |
| CiP (Lei et al., 2023) | SDv1.5 | 79.4 | 62.4 | 54.1 | | | | |
| OURS | SDv1.5 | 90.5 | 80.0 | 70.9 | 64.3 | 81.8 | 89.2 | 94.6 |
| Δ *with the previous state-of-the-art* | | +10.1 | +17.6 | +1.7 | +30.6 | +34.7 | +13.3 | +0.6 |
| *Training **with** real data* | | | | | | | | |
| Baseline *real data only* | - | 93.0 | 86.3 | 79.6 | 81.5 | 89.5 | 93.2 | 97.6 |
| OURS + *real data* | SDv1.5 | 95.1 | 88.2 | 79.9 | 83.5 | 90.3 | 94.0 | 98.9 |
| Δ *with the real data* | | +2.1 | +1.9 | +0.3 | +2.0 | +0.8 | +0.8 | +1.3 |

ablation study on the IN-10 and IN-100 datasets. As illustrated in Tab. 2, every proposed module contributes to remarkable performance improvement. The combination of three modules achieves the best results, which outperform the baseline by 10.7% and 14.0% on IN-10 and IN-100, respectively. Further analysis and details are provided in Apx. G.5.

Table 2: **Ablation study** on proposed improvements with IN-10 and IN-100 Top-1 accuracy. Ticks under each column represent the implementation of the corresponding module in data synthesis. *Finetune* indicates whether finetune Stable Diffusion on target dataset with the augmented distribution matching loss or original diffusion loss Ho et al. (2020).

| Latent Prior Sec. 3.3 | Visual Guidance Sec. 3.2 | Distribution Matching Sec. 3.1 | Finetune | ImageNette | ImageNet100 |
|---|---|---|---|---|---|
| | | | | 79.8 | 66.0 |
| | | | ✓ | 80.8 | 73.3 |
| | ✓ | | ✓ | 82.3 | 74.0 |
| | | ✓ | ✓ | 81.2 | 73.8 |
| | ✓ | ✓ | ✓ | 82.9 | 75.1 |
| ✓ | | | | 88.0 | 76.3 |
| ✓ | | | ✓ | 88.7 | 78.3 |
| ✓ | ✓ | | ✓ | 89.5 | 79.3 |
| ✓ | | ✓ | ✓ | 88.9 | 79.8 |
| ✓ | ✓ | ✓ | ✓ | 90.5 | 80.0 |

## 4.2 AUGMENTING REAL DATA WITH SYNTHETIC DATA

We study whether the synthetic data can serve as dataset augmentation. We compare the models trained with only real data and those trained with both real and synthetic data. As shown in lower part of Tab. 1, we observe improvements across all benchmarks when combining the synthetic and real data. Especially, our synthetic data boosts the performances by 2.1% and 1.9% on IN-10 and IN-100 datasets respectively. This validates that our synthetic data align well with the real data distribution.

## 4.3 SCALING UP SYNTHETIC TRAINING DATA

Besides data distribution alignment, training set cardinality is another key factor influencing the utility of synthetic data identified in Eq. (4). Fortunately, it is easy to synthesize more data for deep generative model. In the experiment, we scale up the synthetic dataset and train the image classifiers using increasing amounts of synthetic data, ranging from $1\times$ to $10\times$ the size of the original real dataset. As shown in Fig. 2, solely scaling

Table 3: Scaling-up synthetic ImageNet-1K.

| Synthetic *vs.* real data size | Real data? | Top-1 | Top-5 |
|---|---|---|---|
| $\times 1$ (1.3M) | | 70.9 | 89.9 |
| $\times 2$ (2.6M) | | 72.9 | 91.1 |
| $\times 5$ (6.4M) | | 74.5 | 92.1 |
| $\times 10$ (13M) | | 76.0 | 93.1 |
| $\times 1$ (1.3M) | ✓ | 79.6 | 94.6 |

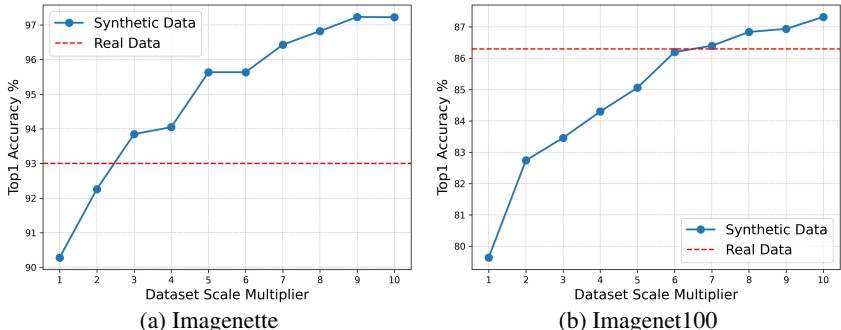

|         (a) Imagenette         |         (b) Imagenet100        |

Figure 2: **Effect of Scaling Up Synthetic Dataset** Top-1 image classification performance with synthetic data only (indicated by blue solid curve) increases with synthetic dataset size, eventually outperforming real data (indicated by red dash line). The horizontal axis represents the amount of synthetic data used as multiples of the original real dataset size.

up the synthetic dataset enables the image classification performance to surpass that of models trained on real data, even **without** trained on it. For the small-scale dataset IN-10, the threshold scale of synthetic data to outperform the real data is only $2.5\times$ the size of the real data. While, the challenge is greater for larger-scale dataset IN-100. To surpass the real data efficacy, synthetic data with $6.5\times$ size is needed on IN-100. As shown in Tab. 3, we also scale up synthetic training images for the large dataset IN-1K, which has 1.3M real samples. The results show that the trend of increasing image classification accuracy along with the increasing synthetic data, still holds, where we achieve 72.82% and 76.00% with 2 × and 10 × ImageNet-1K size, respectively. By scaling up synthetic data, we reduce the gap between real and synthetic data down to 3.6% in top-1 accuracy and more encouraging 1.5% in top-5 accuracy.

## 4.4 GENERALIZATION TO OUT-OF-DISTRIBUTION DATA

We also investigate the Out-of-distribution (OOD) generalization performance on four OOD variants of ImageNet: (1) ImageNet-v2 (IN-v2) (Recht et al., 2019) (2) ImageNet-Sketch (IN-Sketch) (Wang et al., 2019) (3) ImageNet-R (IN-R) (Hendrycks et al., 2021a) (4) ImageNet-A (IN-A) (Hendrycks et al., 2021b). We test the ResNet50 (He et al., 2016) trained with in-distribution real data (i.e. ImageNet-1K training set) or synthetic data (i.e. the generative model is only tuned on ImageNet-1K training set like above experiments) on the OOD test sets.

Table 4: **OOD Generalization Performance.** $\times k$ indicates training solely with the synthetic dataset of scale $k$ times that of real dataset. Top-1 and 5 classification accuracies are reported.

|  | ImageNet-v2 (Recht et al., 2019) | | ImageNet-Sketch (Wang et al., 2019) | | ImageNet-R (Hendrycks et al., 2021a) | | ImageNet-A (Hendrycks et al., 2021b) | |
|---|---|---|---|---|---|---|---|---|
|  | Top-1 | Top-5 | Top-1 | Top-5 | Top-1 | Top-5 | Top-1 | Top-5 |
| *Training **without** real data* | | | | | | | | |
| FakeIt Sarıyıldız et al. (2023) | 43.0 | 70.3 | 16.6 | 35.2 | 26.3 | 45.3 | 3.6 | 15.1 |
| CiP Lei et al. (2023) | 53.8 | 80.5 | 18.5 | 35.5 | 33.6 | 51.1 | 5.2 | 21.7 |
| OURS | 67.2 | 88.0 | 21.9 | 38.2 | 35.4 | 51.9 | 4.5 | 23.5 |
| OURS ×2 | 69.5 | 89.2 | 25.2 | 43.0 | 36.1 | 52.5 | 6.8 | 29.2 |
| OURS ×5 | 71.1 | 90.6 | 27.8 | 46.2 | 39.7 | 56.3 | 9.4 | 34.9 |
| OURS ×10 | 73.0 | 91.4 | 29.2 | 48.1 | 41.0 | 57.5 | 11.3 | 39.4 |
| *Training **with** real data* | | | | | | | | |
| Baseline *real data only* | 74.7 | 92.2 | 28.1 | 45.8 | 39.4 | 54.1 | 8.1 | 34.7 |
| OURS + *real data* | 75.7 | 92.7 | 29.0 | 46.8 | 40.5 | 55.9 | 9.0 | 36.4 |

As illustrated in Tab. 4, we observe that when training with $1\times$ synthetic data only, our method achieves the best generalization performance across three out of four benchmarks, outperforming previous synthesis strategies. When jointly training with real data, our synthetic data further boosts the OOD generalization performance of real data. More importantly, when we scale up the synthetic data, its OOD generalization performance exceeds that of real data, e.g. on ImageNet-Sketch, ImageNet-R and ImageNet-A, even before achieving comparable performance on the in-distribution test set. This further highlights the promising utility of synthetic training data in enhancing OOD generalization.

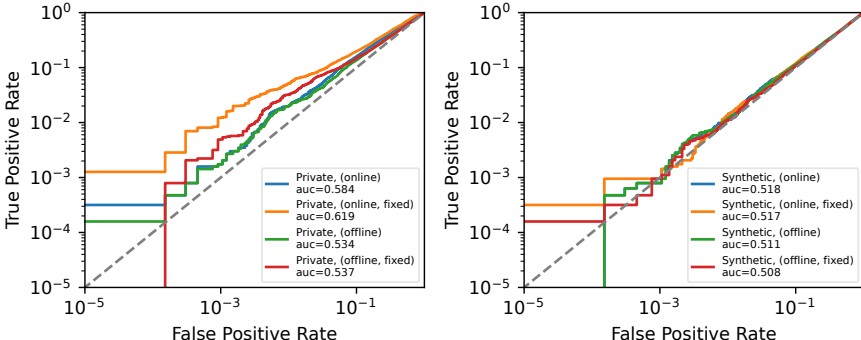

Figure 3: **Membership Inference Attack (MIA) Performance with LiRA.** LiRA achieves a TPR of 0.001% at a low FPR of 0.1% when applied to synthetic data, while the results for private data is 0.01%, which indicates training with synthetic data is much more privacy-preserving.

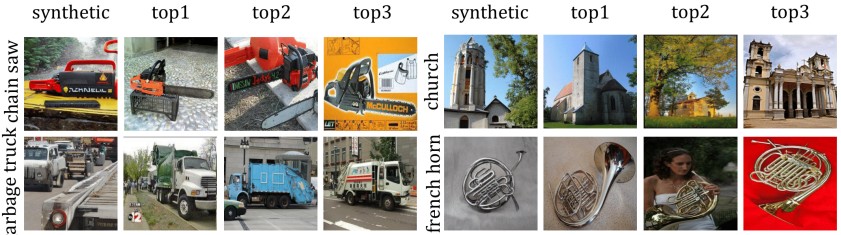

Figure 4: **Visualization on synthetic and retrieved real data with SSCD.** The synthesized data does not exhibit evident copying or memorization.

### 4.5 PRIVACY ANALYSIS

Synthetic training data is a promising solution to privacy-preserving learning. In this subsection, we examine the privacy-preserving nature of our synthetic data from the perspectives of both privacy attacks and visual similarity. Further experiment details are provided in Apx. G.4.

**Membership Inference Attack.** We implement the membership inference attack that enables an adversary to query a trained model and determine whether a specific example is part of the model's training set. Specifically, we follow the state-of-the-art Likelihood Ratio Attack (LiRA) Carlini et al. (2022) and report the MIA results for two training approaches: the privacy training dataset and synthetic data, in the low false-positive rate regime. To account for training duration, we conduct 10 sampling iterations. Concretely, we initially divide the privacy data in IN-10 into two halves: member data and non-member data. We only employ LoRA to finetune the diffusion model on the member data, then generate synthetic data of equal size. Subsequently, we train a ResNet50 on synthetic data. As depicted in Fig. 3, regardless of online attack, offline attack, or fixed variance scenarios, the model trained on synthetic data demonstrates superior defense against MIA.

**Visual Similarity.** We follow the previous works (Zhang et al., 2023; Somepalli et al., 2023) and employ the Self-Supervised Content Duplication (SSCD) (Pizzi et al., 2022) method for content plagiarism detection. We use the ResNeXt101 model trained on the DISC dataset (Douze et al., 2021). For a given query image and all reference images, we infer through the detection model to obtain a 1024-dimensional feature vector representation for each image. By computing the inner product between the query feature and each reference feature, we obtain the similarity between them. Subsequently, we select the the top five reference features with the highest similarity to the query feature. Then, we rank the similarity and present the top 3 images with their corresponding real data in Fig. 4. From the figure, we observe that there are no apparent issues of copy or memorization (Carlini et al., 2023). At least visually, these synthesized images should provide sufficient privacy protection.

## 5 CONCLUSION

In this work, we propose a principled theoretical framework for training data synthesis from a distribution-matching perspective. Based on this, we empirically push the limit of synthetic training data by advancing the state-of-the-art performances over diverse image classification benchmarks. We also demonstrate promising benefits of improving the OOD generalization and privacy preservation performances by training models with our synthetic data.

**Acknowledgement.** This work is funded by National Key R&D Program of China (2021ZD0111102) and NSFC-62306046.

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

## A    MINIMIZATION OF KL-DIVERGENCE OBJECTIVE IN DIFFUSION PROBABILISTIC MODEL

Consider the training objective of the diffusion model, minimization of KL-divergence between the joint distributions of forward and backward process $q$ and $p_{\boldsymbol{\theta}}$ over all time steps:

$$\text{KL}\left(q\left(\boldsymbol{x}_0, \boldsymbol{x}_1, \cdots, \boldsymbol{x}_T\right) \| p_{\boldsymbol{\theta}}\left(\boldsymbol{x}_0, \boldsymbol{x}_1, \cdots, \boldsymbol{x}_T\right)\right) \tag{10}$$

By definition of KL-divergence and the property of Markovian transition, we can further decompose the expression in time (Yang et al., 2022):

$$\text{KL}\left(q\|p_{\boldsymbol{\theta}}\right) = \mathbb{E}_q\left[-\log p\left(\boldsymbol{x}_T\right) - \sum_{t=1}^{T} \log \frac{p_{\boldsymbol{\theta}}\left(\boldsymbol{x}_{t-1} \mid \boldsymbol{x}_t\right)}{q\left(\boldsymbol{x}_t \mid \boldsymbol{x}_{t-1}\right)}\right] + \text{const} \tag{11}$$

Using Jensen's inequality, we can derive:

$$\text{KL}\left(q\left(\boldsymbol{x}_0, \boldsymbol{x}_1, \cdots, \boldsymbol{x}_T\right) \| p_{\boldsymbol{\theta}}\left(\boldsymbol{x}_0, \boldsymbol{x}_1, \cdots, \boldsymbol{x}_T\right)\right) \geq -\log p_{\boldsymbol{\theta}}\left(\boldsymbol{x}_0\right) + \text{const} \tag{12}$$

The objective in training is to maximize the variational lower bound (VLB) of the log-likelihood of the data $\boldsymbol{x}_0$, which is equivalent to minimizing the negative VLB as in Eq. (1).

## B    TRAINING DATA SYNTHESIS FOR SUPERVISED LEARNING

Consider the following formulation of supervised learning (Sutskever, 2023; Cunningham et al., 2008):

$$\text{Pr}_{S \sim D^{|S|}}\left[\text{Test}_D(f) - \text{Train}_S(f) \leq \sqrt{\frac{\log|\mathcal{F}| + \log 1/\delta}{|S|}} \forall f \in \mathcal{F}\right] > 1 - \delta \tag{13}$$

where $D$ denotes the distribution of the whole target dataset and $S$ is a training subset of $D$. $\mathcal{F}$ represents all potential functional space, which can be represented by a neural network with a fixed structure, where each unique combination of parameter values corresponds to a unique $f$. The training and test errors are defined as $\text{Train}_S(f) = \frac{1}{|S|}\sum_{(X_i, Y_i) \in S} L\left(f\left(X_i\right), Y_i\right)$ and $\text{Test}_D(f) = \frac{1}{|D|}\sum_{(X_i, Y_i) \in D} L\left(f\left(X_i\right), Y_i\right)$, respectively. Mathematically, it is evident that when the training loss is minimal and the training set cardinality significantly exceeds the model parameters, the test loss will also be notably low.

We now prove this proposition. Consider the formulation of test error:

$$\text{Pr}_{S \sim D^{|S|}}\left[\text{Test}_D(f) - \text{Train}_S(f) \geq t \text{ for some } f \in \mathcal{F}\right] \leq \sum_{f \in \mathcal{F}} \text{Pr}_{S \sim D^{|S|}}\left[\text{Test}_D(f) - \text{Train}_S(f) \geq t\right]$$

$$= \sum_{f \in \mathcal{F}} \text{Pr}_{\sim D^{|S|}}\left[\frac{1}{|S|}\sum\left(L_i - \mathbb{E}[L]\right) \geq t\right]$$

$$\leq |\mathcal{F}|\exp\left(-2|S|t^2\right) \tag{14}$$

The first step is a straightforward extension of the original formulation. The second step employs Hoeffding's inequality (Hoeffding, 1994). Given a series of independent random variables $X_1, X_2, \ldots, X_n$ bounded by $a_i \leq X_i \leq b_i$, Hoeffding's inequality states:

$$\text{Pr}\left[\sum_{i=1}^{n} X_i - \mathbb{E}\left[\sum_{i=1}^{n} X_i\right] \geq t\right] \leq \exp\left(-\frac{2t^2}{\sum_{i=1}^{n}(b_i - a_i)^2}\right) \tag{15}$$

It bounds the probability that the sum of independent bounded random variables deviates from its expected value beyond a set limit. Consider each $L_i - \mathbb{E}[L]$ as a bounded random variable, capturing the deviation of a specific data point's loss from its expected value. By assuming a bounded loss function $L_i$, which is valid as we can project any unbounded loss function monotonically to a bounded range of $[0, 1]$ through sigmoid function, we obtain the upper bound on the probability of

this disparity surpassing threshold $t$, culminating in the expression $|\mathcal{F}| \exp\left(-2|S|t^2\right)$. This implies the convergence of test error to zero when the size of the training dataset goes to infinite:

$$\lim_{|S|->\infty} \Pr\left[\sup_{f\in\mathcal{F}} \left(\text{Test}_D(f) - \text{Train}_S(f)\right) > \varepsilon\right] = 0 \tag{16}$$

which eventually validates our statement on the implication of training data synthesis.

## C    MAXIMUM MEAN DISCREPANCY (MMD) IN RKHS

The Maximum Mean Discrepancy (MMD) in a Reproducing Kernel Hilbert Space (RKHS) (Aronszajn, 1950) $\mathcal{H}$ provides a metric to measure the difference between two probability distributions.

$$\text{MMD}[\mathcal{F}, p, q] = \sup_{\|\psi_\vartheta\|_{\mathcal{H}} \leq 1} \left(\mathbb{E}_q\left[\psi_\vartheta(\mathcal{T})\right] - \mathbb{E}_p\left[\psi_\vartheta(\mathcal{S})\right]\right) \tag{17}$$

$$\text{MMD}[\mathcal{F}, p, q] = \sup_{\|\psi_\vartheta\|_{\mathcal{H}} \leq 1} \langle \mu_\mathcal{T} - \mu_\mathcal{S}, \psi_\vartheta \rangle_{\mathcal{H}} \tag{18}$$

$$\text{MMD}^2[\mathcal{F}, p, q] = \left[\sup_{\|\psi_\vartheta\|_{\mathcal{H}} \leq 1} \langle \mu_\mathcal{T} - \mu_\mathcal{S}, \psi_\vartheta \rangle_{\mathcal{H}}\right]^2 \tag{19}$$

$$\text{MMD}^2[\mathcal{F}, p, q] = \|\mu_\mathcal{T} - \mu_\mathcal{S}\|_{\mathcal{H}}^2 \tag{20}$$

In Eq. (17), the MMD between distributions $p$ and $q$ is formulated as the supremum of the difference of expectations under these distributions, constrained by the norm of the function $\psi_\vartheta$ in the RKHS. This difference can be further expanded using the inner product in the RKHS, leading to an expression in terms of the mean embeddings of the distributions, as shown in Eq. (18). Squaring the MMD gives us a more interpretable metric, which is the squared distance between the mean embeddings of the two distributions in the RKHS, as depicted in Eq. (19) and Eq. (20). This leads to the derivation from Eq. (5) to Eq. (6) in Sec. 3.1.

## D    DATASET DETAILS

We adopt seven datasets in total across different settings: ImageNette (IN-10) (Howard, 2019), ImageNet100 (IN-100) (Tian et al., 2020), and ImageNet1K (IN-1K) (Deng et al., 2009), CUB (Wah et al., 2011), Cars (Krause et al., 2013), PET (Parkhi et al., 2012), EuroSAT (Helber et al., 2018). The detailed dataset statistics are specified in Tab. 5.

Table 5: Detailed Dataset Statistics.

|  | IN-10 | IN-100 | IN-1K | CUB | Cars | PETS | EuroSAT |
|---|---|---|---|---|---|---|---|
| Dataset Size | 14197 | 141971 | 1419712 | 11788 | 16,185 | 7349 | 27000 |
| Training Data Size | 12811 | 128116 | 1281167 | 5994 | 8144 | 3680 | 21600 |
| Test Data Size | 1000 | 10000 | 100000 | 5794 | 8041 | 3669 | 5400 |
| No. Classes | 10 | 100 | 1000 | 200 | 196 | 37 | 7 |
| Domain | Natural Image | Natural Image | Natural Image | Fine-grain Bird | Fine-grain Cars | Pet Breed | Satellite Image |

## E    TRAINING DATA SYNTHESIS DETAILS

We use Stable Diffusion v1.5 (Rombach et al., 2022) across all benchmarks. Unless specified (i.e. we specifically scale up synthetic data in Sec. 4.3), we generate an equal amount of synthetic data samples as in the corresponding real dataset for both replacing and augmenting real training data. The stable diffusion generation parameters are specified in Tab. 6. We use text prompt mentioned in Sec. 3.2, of the form `"photo of [classname], [Image Caption], [Intra-class Visual Guidance]"` for ImageNet dataset (IN-10, IN-100, IN-1K); for the rest four datasets, we do not use the image caption generated with BLIP2 (Li et al., 2023). Also, all generations use a uniform set of negative prompts `"distorted, unrealistic, blurry, out of frame"`.

Table 6: Hyperparameters used in training data synthesis.

| Model | Sampling steps | Scheduler | Guidance scale | Img_strength | Image size |
|---|---|---|---|---|---|
| Stable Diffusion v1.5 | 30 | UniPC | 2.0 | 0.75 | $512 \times 512$ |

For fine-tuning the Stable Diffusion, we use Low-Rank Adaptation (LoRA) (Hu et al., 2021) with the MMD augmented objective Sec. 3.1 and inject the visual guidance Sec. 3.2. The fine-tuning hyperparameters used are specified in Tab. 7. We use the loss function mentioned in Sec. 3.1, where

Table 7: Hyperparameters used in LoRA fine-tuning

| | Rank | Epoch | Leanring Rate | Batch Size | SNR-$\gamma$ | Image size |
|---|---|---|---|---|---|---|
| Stable Diffusion v1.5 | 300 | | 1e-4 | $8 \times 8$ | 0.75 | $512 \times 512$ |

we augment the MMD distribution matching loss with the simple diffusion model training objective Ho et al. (2020) as in Eq. (21), we choose $\gamma = 0.05$.

$$L_{overall} = L_{Simple} + \gamma L_{MMD} = \frac{1}{|\mathcal{N}|} \sum_{i=1}^{|\mathcal{N}|} || \left( \epsilon - \epsilon_{\theta} \left( x_t, t \right) \right) ||_{\mathcal{H}}^2 + \gamma || \frac{1}{|\mathcal{N}|} \sum_{i=1}^{|\mathcal{N}|} \left( \epsilon - \epsilon_{\theta} \left( x_t, t \right) \right) ||_{\mathcal{H}}^2 \tag{21}$$

## F  DOWNSTREAM IMAGE CLASSIFICATION TRAINING DETAILS

We adopt ResNet-50 (He et al., 2016) through all benchmarks in the image classification task. For all three ImageNet datasets, we train the classifier model from scratch. Follow the training recipe of (Lei et al., 2023) and (Wightman et al., 2021), the training hyperparameters are specified in Tab. 8

## G  EXPERIMENT SETTING DETAILS AND MORE RESULTS

### G.1  IMAGE CLASSIFICATION WITH SYNTHETIC DATA ONLY

We compare with the baselines of previous works on synthetic ImageNet data for supervised image classification. All baselines synthesize equal amounts of data as real datasets. The accuracies reported are from the original baseline papers, except for FakeIt (Sarıyıldız et al., 2023) in CUB (Wah et al., 2011), Cars (Krause et al., 2013), PET (Parkhi et al., 2012), and EuroSAT (Helber et al., 2018). The number reported is based on our implementation of the method with the base prompt `"a photo/image of [classname]"`.

Note that there is a difference in the generation resolution among different methods. In particular BigGAN-deep (Brock et al., 2018), VQ-VAE-2 (Razavi et al., 2019), and CDM (Ho et al., 2022), as reported in (Azizi et al., 2023), use image resolution of $256 \times 256$. CiP (Lei et al., 2023), FakeIt (Sarıyıldız et al., 2023), and our method uses image resolution of $512 \times 512$. Imagen (Azizi et al., 2023) uses image resolution of $1024 \times 1024$. Even though we resize all samples in downstream training to $224 \times 224$, the generation resolution still has influences as reported in (Azizi et al., 2023) potentially due to the generation details and quality. Thus, comparing our method and non-Stable Diffusion baselines can potentially be an unfair comparison. However, our main focus in training data synthesis is in comparison with real data and the comparison between ours and other Stable Diffusion-based method show our framework leads to improved synthetic data informativeness to close the gap between real and synthetic data.

### G.2  AUGMENTING REAL DATA WITH SYNTHETIC DATA

For CUB (Wah et al., 2011), Cars (Krause et al., 2013), and PET (Parkhi et al., 2012) datasets, we use a pre-trained ResNet-50 due to the scarcity of data to train a decent ResNet-50 classifier. For EuroSAT (Helber et al., 2018), ImageNette (IN-10) (Howard, 2019), ImageNet100 (IN-100) (Tian et al., 2020), and ImageNet1K (IN-1K) (Deng et al., 2009) we train the model from scratch. We augment the real data with an equal amount of synthetic data, which effectively raises the amount of training by two.

Table 8: Training Hyperparameters of Different Dataset.

|  | IN-10 | IN-100 | IN-1K | CUB | CARS | PETS | EuroSAT |
|---|---|---|---|---|---|---|---|
| Train Res → Test Res | 224 → 224 | 224 → 224 | 224 → 224 | 448 → 448 | 448 → 448 | 224 → 224 | 448 → 448 |
| Epochs | 200 | 200 | 300 | 200 | 200 | 200 | 200 |
| Batch size | 128 × 8 | 128 × 8 | 512 × 8 | 128 × 8 | 128 × 8 | 128 × 8 | 128 × 8 |
| Optimizer | SGD | SGD | LAMB | SGD | SGD | SGD | SGD |
| LR | 0.1 | 0.1 | 5e-3 | 0.1 | 0.1 | 0.1 | 0.1 |
| LR decay | multistep | multistep | cosine | multistep | multistep | multistep | multistep |
| decay rate | 0.2 | 0.2 | 0.2 | 0.2 | 0.2 | 0.2 | 0.2 |
| decay epochs | 50/100/150 | 50/100/150 | - | 50/100/150 | 50/100/150 | 50/100/150 | 50/100/150 |
| Weight decay | 5e-4 | 5e-4 | 0.02 | 5e-4 | 5e-4 | 5e-4 | 5e-4 |
| Warmup epochs | - | - | 5 | - | - | - | - |
| Label smoothing | - | - | 0.1 | - | - | - | - |
| Dropout | x | x | x | x | x | x | x |
| Stoch. Depth | x | x | 0.05 | x | x | x | x |
| H. flip | - | - | ✓ | - | - | - | - |
| Rand Augment | x | x | 7 / 0.5 | x | x | x | x |
| Mixup alpha | x | x | 0.2 | x | x | x | x |
| Cutmix alpha | x | x | 1.0 | x | x | x | x |
| CE loss → BCE loss | x | x | ✓ | x | x | x | x |
| Mixed precision | ✓ | ✓ | ✓ | ✓ | ✓ | ✓ | ✓ |

## G.3 GENERALIZATION TO OUT-OF-DISTRIBUTION DATA

We train the image classifier on synthetic in-distribution synthetic and real ImageNet-1K and then test on four Out-of-Distribution test sets: (1) ImageNet-v2 (IN-v2) (Recht et al., 2019), which is a different ImageNet test set collected on purpose to eliminate the effect of adaptive overfitting for original ImageNet test set; ImageNet-Sketch (IN-Sketch) (Wang et al., 2019), which contains sketch of ImageNet classes with black and white scheme only; ImageNet-R (IN-R) (Hendrycks et al., 2021a), which contains renditions of ImageNet classes such as art, cartoon, graphics, etc.; and ImageNet-A (IN-A) (Hendrycks et al., 2021b), which contain natural adversarial examples for ResNet models trained with ImageNet-1K training set. In particular, ImageNet-A and ImageNet-R only contain 200 classes as a subset of ImageNet-1K, thus we only consider the output logits of the image classifier over these sub-classes.

## G.4 PRIVACY ANALYSIS

**Membership Attack** In this study, we implement a membership inference attack, which allows an adversary to determine whether a specific example belongs to the training set of a trained model. To achieve this, we adopt the Likelihood Ratio Attack (LiRA) as the state-of-the-art method, as proposed by Carlini et al. Carlini et al. (2022). Our objective is to evaluate the performance of this attack by conducting experiments on two different training approaches: the privacy training dataset and synthetic data. Specifically, we focus on the low false-positive rate regime, where the attack is expected to be more effective. To begin, we train a total of 16 shadow models using random samples from the original IN-10 (ImageNette) dataset. Half of these models are trained on a target point (x, y), while the other half are not. This division allows us to create both IN and OUT models for the target point (x, y). By fitting two Gaussian distributions to the confidences of the IN and OUT models, measured in logit scale, we are able to capture the characteristics of these models. Next, we query the confidence of the target model f on the target point (x, y). Using a parametric Likelihood-ratio test, we compare this confidence value with the distributions obtained from the IN and OUT models. Based on this comparison, we can infer whether the target point (x, y) is likely to be a member of the training set or not. In order to train the shadow models, we utilize the ResNet50 architecture. We always begin by finetuning the diffusion model using the sampled data. Once the finetuning is complete, we generate synthetic data of equal size to the original dataset. Subsequently, we train the shadow models using this synthetic data. As a result, we have a total of 16 finetuned diffusion models and 16 ResNet50 models for our experiments. By implementing the membership inference attack and evaluating its performance on both the privacy training dataset and synthetic data, we aim to provide insights into the vulnerability of trained models to such attacks. The results of our experiments shed light on the effectiveness of the Likelihood Ratio Attack in the low false-positive rate regime and contribute to the ongoing research on model privacy and security.

**Visual Appearance** In this study, we build upon the prior research conducted by Zhang et al. (Zhang et al., 2023) and Somepalli et al. (Somepalli et al., 2023) and utilize the Self-Supervised Content

Duplication (SSCD) method proposed by Pizzi et al. (Pizzi et al., 2022) for content plagiarism detection. SSCD is a self-supervised approach that is based on SimCLR (Chen et al., 2020) and incorporates InfoNCE loss (Oord et al., 2018), entropy regularization of latent space representation, and various data augmentation techniques. For the experiment, we use the ResNeXt101 model trained on the DISC dataset (Douze et al., 2021) provided by the official `https://github.com/facebookresearch/sscd-copy-detection/tree/main` as our detection model. To train our model, we employ the ResNeXt101 architecture and train it on the DISC dataset (Douze et al., 2021). We first extract 1024-dimensional feature vectors for both the query image and all reference images using the trained detection model. These feature vectors are then normalized using L2 normalization. By computing the inner product between the query feature vector and each reference feature vector, we obtain a similarity score, which indicates the degree of similarity between the query and reference images. Based on these similarity scores, we select the top five reference features that exhibit the highest similarity with the query feature. These selected features allow us to retrieve the corresponding images from the dataset. To further evaluate the similarity between the query and reference images, we rank the similarity scores and present the top 3 images along with their corresponding real data.

## G.5 FURTHER ABLATION STUDY ANALYSIS

In Sec. 4.1, we conducted an ablation study on three key techniques: (1) Latent Prior, (2) Visual Guidance, and (3) Distribution Matching with MMD loss. Note that, *Visual Guidance* and *Distribution Matching* require finetuning on Stable Diffusion with either simple diffusion model loss Ho et al. (2020) if Distribution Matching not adopted or otherwise augmented distribution matching loss. *Latent Prior* can be adopted with or without finetuning on Stable Diffusion. This section provides a more detailed analysis of the effectiveness of each module.

**Latent Prior**    The application of Latent Prior demonstrates significant performance improvements. Specifically, implementing Latent Prior results in top-1 accuracy enhancements of 8.2% and 10.3% on ImageNette and ImageNet-100, respectively, when compared to the baseline. This underscores the efficacy of Latent Prior in our framework. However, even without Latent Prior, Visual Guidance and Distribution Matching Loss also contribute to performance enhancements. Without Latent Prior, these methods lead to improvements of 1.1% and 1.8% on ImageNet-10 and ImageNet-100, respectively. When integrated with Latent Prior, the improvement further increases to 2.5% and 3.7% on these datasets.

**Visual Guidance**    For Visual Guidance, we employ the "[intra-class Visual Guidance]" token in addition to the base prompt template "photo of [classname], [Image Caption]". We achieve substantial performance gains with the inclusion of Visual Guidance, leading to improvements of 1.5% and 0.7% on ImageNet-10, and 0.6% and 0.2% on ImageNet-100, respectively, when combined with the Latent Prior. This demonstrate the benefit of having detailed image features along with textual description for informative training data synthesis.

**Distribution Matching Loss**    Motivated by the known limitations of diffusion loss's looseness, as discussed in Sec. 3.1, we further include MMD loss to have a tigher bound over the distribution discrepancy between real and synthetic data. The results demonstrate that the addition of MMD loss brings about consistent improvements. Specifically, on top of the vanilla diffusion loss, the MMD loss contributes an increase of 0.4%/0.5% on ImageNette and ImageNet-100. When combined with Visual Guidance, these improvements are 0.6%/1.1%, and with both Latent Prior and Visual Guidance, they are 1.0%/0.7%, respectively.

## G.6 MULTI-SEED EXPERIMENT

To examine the stability of our proposed method, we deliver a three-seed multi-run experiment on the ImageNet (IN-10, 1N-100, IN-1K) dataset, following the same training recipe as in Tab. 6. As demonstrated in Tab. 9, we observe that for small-scale dataset (IN-10), the variability is larger compared to larger datasets (IN-100, IN-1K), but for all settings, variation remains within an acceptable range and maintains a substantial margin against the baseline method.

Table 9: **Multi-Seed Experiment Results**. Top-1 Classification Accuracy across various datasets (IN-10, IN-100, IN-1k) over multiple runs. *Average* shows the mean and standard deviation of results of multiple runs.

|         | IN-10          | IN-100         | IN-1K          |
|---------|----------------|----------------|----------------|
| Run 1   | 90.5           | 80.0           | 70.9           |
| Run 2   | 91.2           | 79.9           | 70.6           |
| Run 3   | 90.7           | 80.2           | 70.9           |
| Average | $90.8 \pm 0.29$ | $80.0 \pm 0.12$ | $70.8 \pm 0.14$ |

### G.7 MORE COMPLEX HUMAN FACE DATASET

To further explore the applicability of our proposed framework, we conduct experiments on more complex tasks involving human faces. The challenges involved with the task on the human face usually come with more fine-grained facial attributes and natural demographic bias with spurious correlation in data distribution. Thus, we leverage the CelebA Liu et al. (2015) dataset under a general facial attribution classification task. Consistent with other experiments, we synthesize training data with Stable Diffusion 1.5 and train a ResNet50 from scratch for 30 epochs.

**Facial Attribute Classification**   One of the primary tasks related to human face recognition is attribute classification. We conduct this on the CelebA dataset, with three different attributes for prediction. For synthesizing training data, the `"A [Attribute Name] person"` serves as the base prompt template. We select a subset of CelebA dataset and formulate three binary classifications of *Smiling*, *Attractive*, and *Heavy Makeup*.   As shown in Tab. 10, for concert facial attribute like

Table 10: CelebA Facial Attribute Classification Result.

|           | Smiling | Attractive | Heavy Makeup |
|-----------|---------|------------|--------------|
| Real Data | 89.65   | 86.22      | 85.60        |
| Baseline  | 76.90   | 65.11      | 73.35        |
| OURS      | 86.93   | 74.74      | 82.55        |

*Smiling* and *Heavy Makeup*, we observe synthetic data from our framework can represent facial attribute levels close to the training data. For more subtle concepts like *Attractive*, the distribution shift between synthetic data and real data is larger, leads to slightly less optimal results. In all cases, our synthetic data outperform the baseline by a considerable margin. This validates the capability of our framework to adapt to fine-grained facial features.

**Future work in More Complex Face Recognition**   While we primarily explore the utility of synthetic data in capturing general facial features, delving into the domain of individual human face recognition presents a considerably more challenging task. This complexity arises from the extreme scarcity of individual-specific data and the heightened need for privacy preservation in such contexts. Recognizing the significance and complexity of this area, we identify it as an important avenue for future work.

## H FURTHER DISCUSSION

### H.1 SIGNIFICANT OF CLASS-LIKELIHOOD MATCHING IN FINE-GRAINED CLASSIFICATION

In Sec. 4.1, we observe a more significant advantage of our synthetic data over fine-grain classification tasks (i.e. CUB (Wah et al., 2011), Cars (Krause et al., 2013)) than natural image classification (i.e ImageNet (Deng et al., 2009)) in comparison to baseline. This is due to the requirement of more detailed decision boundaries among classes. This highlights the importance of the class-likelihood alignment $q(y|\boldsymbol{x}) = p_{\boldsymbol{\theta}}(y|\boldsymbol{x})$. This also indicates solely increasing in the capacity of the generative model in synthesizing high-quality images is insufficient to handle hard image classification without explicit alignment between the joint distribution between image and annotation.

## H.2 BETTER OUT-OF-DISTRIBUTION GENERALIZATION PERFORMANCE

In the assessment of OOD generalization performance Sec. 4.4, we find that with scaling-up ImageNet-1K performance, although the in-distribution performance of training with synthetic data is still inferior to real data, the OOD generalization performance already outperforms the real counterpart at an earlier breaking point (i.e. in ImageNet-R (IN-R) (Hendrycks et al., 2021a) and ImageNet-A (IN-A) (Hendrycks et al., 2021b)). While we synthesize training data to reproduce the target data distribution, the superior OOD generalization for models trained on synthetic data indicates a greater potential for synthetic data in strengthing the OOD generalization performance.

