# OpenReview forum: "Real-Fake: Effective Training Data Synthesis Through Distribution Matching"
_ICLR.cc/2024/Conference — ICLR 2024 poster_

### Official Review · Reviewer_wBAQ · 2023-10-31

**Soundness:** 1 poor
**Presentation:** 3 good
**Contribution:** 2 fair
**Rating:** 5
**Confidence:** 4

**Summary:**

The authors introduce a distribution matching (MMD) framework and implements it within the Stable Diffusion to improve data synthesis. In particular, feature distribution alignment, conditional visual guidance and latent prior initialization are introduced to finetune the Stable Diffusion.  Experiments demonstrate the effectiveness of their synthetic data across image classification, OOD generalization and privacy preservation tasks.

**Strengths:**

1. The paper is easy to follow.
2. Experiments on various tasks are provided and the results appear promising.

**Weaknesses:**

1. The overall idea of Maximum Mean Discrepancy (MMD) for learning data distribution is not novel, which has been explored before, like [1].
2. The derived MMD loss based on diffusion model (Eq. (7)) is the same as the simplified diffusion model loss (Ho et al. 2020). This means the proposed loss does not bring anything new for the current diffusion model.
3. The effectiveness of the proposed MMD loss (Eq. (7)) is doubtful. From Table 2, the performance gain brought by MMD is not significant compared to finetuning only.
4. The promising performance gains are significantly attributed to the latent prior (Table 2). Given that the latent prior improvement is not introduced by this work but has been already explored by the literature (Section 3.3), the contribution of this work is marginal.

Minor:
1. $\epsilon_0$ is not defined in Eq. (7).

**Questions:**

See the weakness above. Other questions:

1. For the finetune only in Table 2, does it mean finetuning the Stable Diffusion with original losses on the target datasets?
2. For the latent prior only in Table 2, does it mean simply adopting the proposed prior for the pretrained Stable Diffusion to synthesiz the target images?

---

> ### Author Response · Authors · 2023-11-16
>
> Dear Reviewer wBAQ,
>
>
>
>
> Thank you for your valuable feedback and constructive comments. We appreciate you recognizing our work is easy to follow and results appear promising. We now address your concerns and questions below.
>
>
>
>
> **Q1: Maximum Mean Discrepancy (MMD) for learning data distribution is not novel, which has been explored before.**
>
>
> **A1:** We kindly note the main idea of our work is not leveraging MMD loss as a standalone technique for learning data distribution but to formulate the training data synthesis as a distribution matching problem (Eq. 3 and Eq. 4). Our specific implementation of this framework on latent diffusion model involves using distribution matching loss (MMD) **along with latent prior initialization and conditional visual guidance**, integrated in Stable Diffusion for synthesizing informative training data. The significant performance improvements brought by our method empirically indicate the effectiveness of the proposed framework. We have also **validated the importance of all three components** of our method. We have revised our manuscript to emphasize and clarify our main idea.
>
>
>
>
> **Q2: The derived MMD loss based on diffusion model (Eq. (7)) is the same as the simplified diffusion model loss (Ho et al. 2020).**
>
>
> **A2:** We would like to clarify that our derived MMD loss is **not** identical to the diffusion model loss presented by Ho et al. (2020). Their simplified diffusion model loss is a mean square error calculated on top of the predicted noise and added noise whereas the MMD loss, Eq. (7), is the L2-norm of the averaged difference between the predicted noise and added noise.
>
>
> $$
> L_{DM} := ||\frac{1}{|N|} \sum_{i=1}^{|N|} (\epsilon - \epsilon_{\theta}(x_t, t))||^2 \leq
> \frac{1}{|N|} \sum_{i=1}^{|N|} ||(\epsilon - \epsilon_{\theta}(x_t, t))||^2 = L_{Diffusion}.
> $$
>
>
>
>
>
>
> **The diffusion model loss is an upper bound of the MMD loss** by Jensen’s inequality. The derived MMD loss is thus a tighter bound on the distribution discrepancy between real and synthetic data under the MMD measure. We have revised the relevant sections to better clarify this distinction.
>
>
>
>
> [1] Jonathan Ho, Ajay Jain, and Pieter Abbeel. Denoising diffusion probabilistic models. Advances in neural information processing systems 33:6840–6851, 2020.
>
>
>
>
> **Q3: From Table 2, the performance gain brought by MMD is not significant compared to finetuning only.**
>
>
> **A3:** As mentioned above, we show that the diffusion loss (Ho et al. 2020) is an upper bound of the MMD loss. Therefore, finetuning with the diffusion model loss still aligns with our distribution matching framework for training data synthesis. Our motivation (Section 3.1) to further include MMD loss rather than simply diffusion model loss is due to the known looseness problem of diffusion loss (Kingma et al. 2021). The results in Table 2 also show that the extra MMD loss can bring notable improvement in some settings, which are selected in the following table for illustration. MMD loss leads to consistent improvements of 0.6/1.1%, and 1.0/0.7%  on ImageNette/ImageNet100, on top of vanilla diffusion loss, accordingly. We revised our manuscript to clarify it and provided further analysis in Appendix G.5.
>
>
>
>
> | Latent Prior | Visual Guidance | Distribution Matching | Finetune | ImageNette | ImageNet100 |
> |--------------|-----------------|-----------------------|----------|------------|-------------|
> |              | ✓               |                       | ✓        | 82.3       | 74.0        |
> |              | ✓               | ✓                     | ✓        | 82.9       | 75.1        |
> | -  |         -        |          -             |     -     | $\Delta$ + 0.6 | $\Delta$ + 1.1 |
> | ✓            | ✓               |                       | ✓        | 89.5       | 79.3        |
> | ✓            | ✓               | ✓                     | ✓        | 90.5       | 80.0        |
> |     -             |        -         |             -          |     -     | $\Delta$ + 1.0 | $\Delta$+ 0.7 |
>
>
>
>
> [2] Diederik Kingma, Tim Salimans, Ben Poole, and Jonathan Ho. Variational diffusion models. Advances
> in neural information processing systems, 34:21696–21707, 2021.

---

> > ### Author Response · Authors · 2023-11-16
> >
> > **Q4: The promising performance gains are significantly attributed to the latent prior (Table 2). Given that the latent prior improvement is not introduced by this work but has been already explored by the literature (Section 3.3), the contribution of this work is marginal.**
> >
> >
> > **A4:** We would like to note that while previous studies have primarily utilized latent prior for general image synthesis, as discussed in Sec. 3.3, its application in training data synthesis remains relatively under-explored. **Our approach integrates latent prior within a theoretical framework, offering a novel perspective in this field.** Furthermore, compared with the diffusion inversion-based method Zhou et al. (2023), our latent prior (Section 3.3) is computationally efficient and can scale up to large datasets such as ImageNet-1K, a capability not demonstrated by the related work. Additionally, our framework’s efficacy is not confined to the latent prior. **The other two components – Visual Guidance and Distribution Matching Loss – also contribute to remarkable performance improvements.** As shown in the selected table below (from Table 2), without the latent prior, the other two components result in the enhancements of **2.1% and 1.8%** on ImageNette and ImageNet-100, respectively. When integrated with the latent prior, the other two components further boost the performance by **0.8% and 1.7%** on the same datasets. We polished our manuscript to clarify these points in Appendix G.5.
> >
> >
> >
> >
> > | Latent Prior | Visual Guidance | Distribution Matching | Finetune | ImageNette | ImageNet100 |
> > |--------------|-----------------|-----------------------|----------|------------|-------------|
> > |              |                 |                       | ✓        | 80.8       | 73.3        |
> > |              | ✓               | ✓                     | ✓        | 82.9       | 75.1        |
> > |      -        |        -         |             -          |   -       |$\Delta$ + 2.1 |$\Delta$ +1.8 |
> > | ✓            |                 |                       |  ✓        | 88.7     | 78.3        |
> > | ✓            | ✓               | ✓                     | ✓        | 90.5       | 80.0        |
> > |      -        |        -         |             -          |   -       |$\Delta$ + 2.5 |$\Delta$ +3.7 |
> >
> >
> >
> >
> >
> >
> >
> >
> > [3] Yongchao Zhou, Hshmat Sahak, and Jimmy Ba. Training on thin air: Improve image classification
> > with generated data. arXiv preprint arXiv:2305.15316, 2023.
> >
> >
> >
> >
> > **Q5: $\mathbf{\epsilon_{0}}$ not defined in Eq. (7).**
> >
> >
> >
> >
> > **A5:** Thank you for pointing out this problem. $\mathbf{\epsilon_{0}}$ represents the added noise $\mathbf{\epsilon}$ in the forward process of the diffusion model at each time step. We have corrected it in Eq.(7) in Section 3.1.
> >
> >
> >
> >
> > **Q6: For the finetune only in Table 2, does it mean finetuning the Stable Diffusion with original losses on the target datasets?**
> >
> >
> >
> >
> > **A6:** Yes, It refers to finetuning the Stable Diffusion with the original loss from Ho et al. (2020) on the target datasets. We have clarified it in the updated Table 2.
> >
> >
> >
> >
> > **Q7: For the latent prior only in Table 2, does it mean simply adopting the proposed prior for the pre-trained Stable Diffusion to synthesize the target images?**
> >
> >
> >
> >
> > **A7:** Yes, the “Latent Prior” experiment setting only refers to adopting the prior for pre-trained Stable Diffusion without finetuning. We clarified it in the updated Table 2 and also Appendix G.5 for clarity.

---

> ### Comment · Reviewer_wBAQ · 2023-11-17
> **Thanks for the reponses.**
>
> 1. I think it is trivial to formulate training data synthesis as a problem of learning data distribution, in particular a joint data distribution $q(x, y)$. This type of joint data distribution learning has been well established in the literature, e.g., [1]. The author emphasizes the significance of this contribution, with which I do not agree. Implementing the joint distribution learning through feature distribution alignment and conditional distribution alignment is a standard procedure, not novel.
>
> 2. From the description in Section 3.3, the component of latent prior initialization is not derived from the joint distribution learning framework but rather from the literature on the diffusion model. Therefore, I am not convinced by the claim in A4 that the proposed approach “integrates latent prior within a theoretical framework, offering a novel perspective in this field”.
>
> 3. I was misled by Eq. (21) before, erroneously thinking that the work employed the upper bound of $L_{DM}$ as the objective for DM. Thanks for the clarification in A2. It is clear to me now. I suggest polishing Eq. (21) to avoid confusion.
>
> 4. The superiority of MMD loss over original diffusion model loss is not well verified. The authors claim that "Our motivation (Section 3.1) to further include MMD loss rather than simply diffusion model loss is due to the known looseness problem of diffusion loss (Kingma et al. 2021)." Then a comparison of the MMD loss solely with the simplied diffusion model loss should be included.
>
> 5. Given the responses, now I believe deriving MMD loss under the diffusion model introduces some novelty. The authors may consider re-organize the work for delivering the contributions of this work. I think the current emphasis on a principled theoretical framework from the joint distribution matching perspective overstate the contributions of the work.
>
>
> [1] Li, C., Xu, T., Zhu, J., & Zhang, B. (2017). Triple generative adversarial nets. Advances in neural information processing systems, 30.

---

> > ### Author Response · Authors · 2023-11-19
> >
> > Dear Reviewer wBAQ,
> >
> > Thank you for your further feedback and suggestions. We are glad to see that our response **clarifies your misunderstanding of the objective function, and now you “believe deriving MMD loss under the diffusion model introduces some novelty”.** We sincerely hope that you can improve the rating based on these.
> >
> > Here, we would like to address your concern by further clarifying the novelty and contribution of our work:
> > 1. Novelty: It would be fair to discuss the novelty of our method in the context of the training data synthesis problem instead of focusing on some specific design that has been explored in the literature in related area. We fully understand your concern about the novelty and contribution. However, we still believe that the explicit introduction and contextualization of the distribution matching framework in training data synthesis, along with our highly effective method, is meaningful and beneficial to the area.
> >
> > 2. Contribution: This paper contributes an effective method with three components based on a theoretical framework for synthesizing informative training samples. Experiments have proven that we advance the state-of-the-art by a large margin, i.e., **10% ~ 30% improvements** using the same open-source Stable Diffusion model. We further experimentally verify that our synthetic training data can protect data privacy and improve models’ out-of-distribution performance.
> >
> >
> > Thank you again for your active and constructive feedback! We have revised Eq.21 and cited the literature in related area and we will further polish the paper based on your suggestions.
> >
> > All Authors

---

> > > ### Comment · Reviewer_wBAQ · 2023-11-20
> > > **I increase the score, but still not vote for acceptance.**
> > >
> > > I increase the score for the novelty of MMD loss under the context of diffusion models. However, I still maintain my opinions regarding the points #1, 2, 4, 5 in my last responses.

---

> > > > ### Author Response · Authors · 2023-11-20
> > > >
> > > > Dear Reviewer wBAQ,
> > > >
> > > >
> > > > We sincerely thank you for raising your rating. We will keep polishing the paper based on your valuable feedback.
> > > >
> > > >
> > > > All Authors

---

### Official Review · Reviewer_BpUK · 2023-11-01

**Soundness:** 3 good
**Presentation:** 4 excellent
**Contribution:** 2 fair
**Rating:** 5
**Confidence:** 4

**Summary:**

This paper proposes a principled theoretical framework for training data synthesis from a distribution matching perspective. The key idea is to frame training data synthesis as matching the joint distribution of data and labels between real and synthetic datasets. This involves two aspects: (1) matching the marginal data distribution, and (2) matching the conditional label distribution given data.

Based on this framework, the authors improve upon the state-of-the-art Stable Diffusion model for image synthesis through three strategies: (1) adding an MMD loss to better align feature distributions, (2) incorporating visual cues along with text prompts for conditioned sampling, and (3) using a informative prior for latent diffusion sampling.

Experiments demonstrate improved image classification performance using the proposed synthetic data, outperforming previous methods. Advantages are shown in replacing real data, augmenting real data, scaling up synthetic data, and for OOD generalization and privacy preservation.

**Strengths:**

* The paper offers a theoretical perspective on training data synthesis, formalizing the objective as distribution matching between real and synthetic datasets. The framework clearly decomposes the overall problem into marginal data distribution matching and conditional label distribution matching. This exposes the different requirements.
* The proposed improvements are well-motivated based on aligning with the principles of the theoretical framework.
* Thorough experiments validate the utility of the proposed synthetic data on diverse image classification benchmarks. Gains are shown over state-of-the-art baselines. The work highlights promising advantages of synthetic data for OOD generalization and privacy preservation.

**Weaknesses:**

* The improvment over Imagen seems limited. The comparisons between the proposed method and Imagen are limited to ImageNet-1K, making the superiority unclear on other datasets.

* This work utilizes the prompt in format of "photo of [classname], [Image Caption], [Intra-class Visual Guidance]". Are there any other comparative methods that also use the same "[Image Caption]"? Whether the improvment are mainly from "[Image Caption]" instead of "[Intra-class Visual Guidance]"?

**Questions:**

see weakness

---

> ### Author Response · Authors · 2023-11-16
>
> Dear Reviewer BpUK,
>
>
> Thank you for your insightful comments on our paper. We appreciate that you consider our work as a well-motivated theoretical framework, and recognize the comprehensive experiments with gains over the state of the art and promising advantages for OOD generalization and privacy preservation. We now address your concerns below.
>
>
>
>
> **Q1: Improvement over Imagen is limited. Only the comparison on ImageNet-1K is provided.**
>
>
> **A1:** Thank you for raising this point regarding the comparison to Imagen results (Azizi et al. 2023). We would like to kindly highlight several key aspects to consider Imagen (Saharia et al. 2022) in this context.
>
>
> Firstly, Imagen is a **closed-source** model trained on **private data**, which makes it impossible for us to obtain more experiment results of Imagen to compare except their reported ones on ImageNet-1K dataset in (Azizi et al. 2023).
>
>
> Secondly, the Imagen, trained on **private data**, involves a **more powerful generative model than Stable Diffusion** and is **fine-tuned with full parameters** on ImageNet-1K, which is computationally unaffordable by general researchers. In contrast, our approach leverages the open-source Stable Diffusion model coupled with a low-budget tuning (LoRA) and synthesis framework.
>
>
> Thirdly, despite the resource constraints, our method still achieves **notable performance improvements (1.7%)** on the very challenging large-scale dataset ImageNet-1K, underlining the effectiveness of our techniques.
>
>
> Last but not least, with a fair comparison with other baselines using the same generative model backbone, Stable Diffusion, our synthesis framework brings significant (10% ~ 35%) performance improvements on other datasets.
>
>
>
>
> [1] Chitwan Saharia, William Chan, Saurabh Saxena, Lala Li, Jay Whang, Emily L Denton, Kamyar Ghasemipour, Raphael Gontijo Lopes, Burcu Karagol Ayan, Tim Salimans, et al. Photorealistic text-to-image diffusion models with deep language understanding. Advances in Neural Information Processing Systems, 35:36479–36494, 2022
>
>
> [2] Shekoofeh Azizi, Simon Kornblith, Chitwan Saharia, Mohammad Norouzi, and David J Fleet. Synthetic data from diffusion models improves imagenet classification. arXiv preprint arXiv:2304.08466, 2023
>
>
>
>
> **Q2: Are there any other comparative methods that also use the same "[Image Caption]"? Whether the improvement are mainly from "[Image Caption]" instead of "[Intra-class Visual Guidance]"?**
>
>
> **A2:** Thanks for the question. Actually, the CiP (Lei et al. 2023) method generates images based on the "[Image Caption]". Specifically, they use the template “A photo of {class name}, {image caption}” to generate prompts. We provided a comparison to their results in Table 1 in the submission. Our method outperforms the baseline by 10%, 18%, and 17% on ImageNette, ImageNet-100 and ImageNet-1k datasets respectively.
> Furthermore, our ablation study in Table 2 has proven the importance of "[Intra-class Visual Guidance]" component. We further clarify it by showing the following selected results from Table 2. As shown, "[Intra-class Visual Guidance]" brings 1.5% and 0.7% performance improvements on ImageNette and ImageNet100 respectively, when cooperating with “Distribution Matching” components. To clarify it, we revised our manuscript and provided further analysis in Appendix G.5.
>
>
>
>
> | Latent Prior | Visual Guidance | Distribution Matching | Finetune | ImageNette | ImageNet100 |
> |--------------|-----------------|-----------------------|----------|------------|-------------|
> |              |                 |                       | ✓        | 80.8       | 73.3        |
> |              | ✓               |                       | ✓        | 82.3       | 74.0        |
> |      -        |        -         |             -          |   -       |$\Delta$ + 1.5 |$\Delta$ +0.7 |
>
>
>
>
> [3] Shiye Lei, Hao Chen, Sen Zhang, Bo Zhao, and Dacheng Tao. Image captions are natural prompts for text-to-image models. arXiv preprint arXiv:2307.08526, 2023.

---

### Official Review · Reviewer_8NDL · 2023-11-05

**Soundness:** 3 good
**Presentation:** 4 excellent
**Contribution:** 4 excellent
**Rating:** 6
**Confidence:** 3

**Summary:**

This paper studies the problem of synthesizing data which better matches the distribution of real data, and further improve the performance of model training. It comes up with a framework containing two parts of emphasizing the synthetic data distribution. Based on the experimental and theoretical analysis, the performance improves by a significant number on different tasks.

**Strengths:**

1. The study problem is interesting. As the synthetic data is easier to obtain and the real data often comes with privacy concerns, it is useful to understand how synthetic data can contribute, as well as the training process;
2. The proposed framework is sound and easy to understand;
3. The experiments are extensive, and different types of tasks are considered.

**Weaknesses:**

1. It seems there is only 1 run on all the experiments. It is beneficial to include more than 1 repeats for the experiments to exclude the confounders;
2. The experiments are extensive, and I'd also like to see how the proposed framework work on more complex tasks, for example tasks on human faces;
3. Some figures can be improved, for example, Fig. 3, the text overlaps with the curves;

**Questions:**

Please refer to the previous sections

---

> ### Author Response · Authors · 2023-11-16
>
> Dear Reviewer 8NDL,
>
>
>
>
> Thank you for your insightful feedback and constructive comments on our manuscript. We appreciate that you find the study problem interesting, the proposed framework sound, and the experiments extensive. We now address your concern below:
>
>
>
>
> **Q1: More rounds of experiments**
>
>
> **A2:** Thank you for reminding us of this problem. We have included additional multi-runs for ImageNet (IN-10, 1N-100, 1N-1K) experiments as shown in the table below, where we find the standard deviation of the three-seed experiment on all scales stays within a small range. This validates the stability of our method. We have added these results in our revised manuscript in Appendix G.6.
>
>
>
>
> Multi-Seed Experiment Results: Top-1 Classification Accuracy across various datasets (IN-10, IN-100, IN-1k) over multiple runs. *Average* shows the mean and standard deviation of multiple run results.
>
>
>
>
> |       | IN-10      | IN-100     | IN-1k      |
> |-------|------------|------------|------------|
> | Run 1 | 90.5       | 80.0       | 70.9       |
> | Run 2 | 91.2       | 79.9       | 70.6       |
> | Run 3 | 90.7       | 80.2       | 70.9       |
> | Average | 90.8 ± 0.29 | 80.0 ± 0.12 | 70.8 ± 0.14 |
>
>
>
>
> **Q1: Application to more complex task, for example tasks on human face**
>
>
> **A2:** Thank you for suggesting testing our framework on more complex tasks. We conducted additional experiments with a subset of the human face dataset CelebA (Liu et al. 2015) that included a general facial attribution classification task. As shown in the table below, the training data synthesized by our framework adapts well to various facial attributes, with close performance to the real data and notable improvement against the FakeIt baseline. We leave more complex face recognition tasks as future work. We have added the experiment settings, results, and discussion in our revised manuscript in Appendix G. 7.
>
>
>
>
> **CelebA Facial Attribute Classification Results on Three Attributes**
> |               | Smiling | Attractive | Heavy Makeup |
> |---------------|---------|------------|--------------|
> | Real Data     | 89.65   | 86.22      | 85.60        |
> | Baseline          | 76.90  |65.11      | 73.35        |
> | OURS          | 86.93   | 74.74      | 82.55        |
>
>
> [1] Ziwei Liu, Ping Luo, Xiaogang Wang, and Xiaoou Tang. Deep learning face attributes in the wild. In Proceedings of International Conference on Computer Vision (ICCV), 2015.
>
>
> **Q3: Some figures can be improved**
>
>
> **A3:**  We appreciate your helpful suggestions on the figures, especially Fig. 3. We have polished Figure 3 to ensure that all elements are clearly visible and the information is conveyed effectively. We also polished other figures in the manuscript, e.g., adjusting the colors in Figure 1.

---

### Official Review · Reviewer_WEFi · 2023-11-08

**Soundness:** 4 excellent
**Presentation:** 4 excellent
**Contribution:** 3 good
**Rating:** 8
**Confidence:** 3

**Summary:**

The paper proposes a novel diffusion-based approach for generation of synthetic training data. The key innovations of the method are three-fold. Firstly, an MMD-based distribution matching objective between the synthetic and target datasets is employed as fine-tuning step for the diffusion model. Second, visual guidance in the form of intra-class mean features from CLIP plus image captions and class names are used to condition the diffusion model. Lastly, a VAE is employed to obtain the latent codes of real samples for initializing the latent prior of the diffusion model. Experimental evaluations validate the efficacy of the proposed approach across a range of settings and datasets.

**Strengths:**

**Clarity.** The writing, figures, and tables are exceptionally clear.

**Originality.** The paper contains three innovations: feature distribution alignment via MMD, conditioned visual guidance, and VAE-based latent prior initialization. While not algorithmically revolutionary, these innovations are well-motivated and highly effective.

**Quality.** The paper provides interesting theoretical justifications for the proposed innovations and a thorough evaluation of the proposed method, including adequate comparisons to the SoTA and a carefully conducted ablation study.

**Significance.** The proposed approach outperforms the SoTA by huge margins (10%, 17%, 30%, etc.) across a range of datasets for both in and out of distribution data.

**Weaknesses:**

Details of privacy analysis are missing.

**Questions:**

Why are so many of the cells in table 1 empty?

---

> ### Author Response · Authors · 2023-11-16
>
> Dear Reviewer WEFi,
>
>
>
>
> Thank you for your insightful review and the affirmative evaluation of our work. We appreciate that you recognize the good clarity, originality, quality, and significance of our paper. We now address your concern below:
>
>
>
>
> **Q1: Details of privacy analysis are missing.**
>
>
> **A1:**  Thank you for pointing out the insufficiency of details. We addressed this by updating a comprehensive privacy analysis details in Appendix G.4 in the revised manuscript.
>
>
>
>
> **Q2: Missing Entry in Tab1**
>
>
> **A2:** We would like to kindly note a few reasons for the empty entries in Table 1. Firstly, direct evaluations against the Imagen model on diverse datasets are challenging due to its closed-source nature. Secondly, relatively older baselines in Table 1 have been surpassed by more advanced methods, making them less competitive for current benchmarks. Therefore, we focused our evaluations on the large-scale ImageNet-1K dataset, considering it the most representative and classical benchmark for evaluation in this domain. Also, regarding the benchmarking on other datasets, we compare with the current state-of-the-art *FakeIt*. It delivers a fair comparison given the same generative model backbone and demonstrates significant performance improvement.

---

### Author Response · Authors · 2023-11-16

Dear Reviewers and ACs:


Thank you very much for your insightful reviews and constructive comments which help improve our manuscript. We have carefully taken all the suggestions and polished the submission.


Specifically, we have made the following changes:


**In the main paper:**
1. Added scaling-up experiment of 10 $\times$ ImageNet-1k which further narrows the gap between the pure synthetic and real training data performance.
2. Corrected the notation in Eq.(7) and Eq.(21) as suggested by Reviewer wBAQ
3. Modified Abstract and Sections 1 to further clarify the motivation of distribution matching framework and contribution of proposed training data synthesis strategy as suggested by Reviewer wBAQ.
4. Modified Section 3.1 to further clarify the motivation of latent prior initialization as suggested by Reviewer wBAQ.
5. Modified Section 4.1 to clarify the ablation study setting as suggested by Reviewer wBAQ.
6. Improved the display of Figure 1 and Figure 3 as suggested by Reviewer 8NDL
7. Fixed some typos.


**In the supplementary material:**
1. Added more implementation details of the privacy experiment in Appendix G.4 as suggested by Reviewer WEFi.
2. Added further ablation study analysis on the conditional visual guidance in Appendix G.5. as suggested by Reviewer BpUK.
3. Added further ablation study analysis on the latent prior and distribution matching objective in Appendix G.5. as suggested by Reviewer wBAQ.
4. Added more results with mean and standard deviations of multi-seed ImageNet (IN-10, IN-100, IN-1K) classification in Appendix G.6 as suggested by Reviewer 8NDL.
5. Added experiments and discussion on human face dataset in Appendix G.7 as suggested by Reviewer 8NDL.



Thank you again!
Best Regards, All Paper 1261 authors.

---

> ### Author Response · Authors · 2023-11-20
>
> Dear Reviewers and ACs:
>
>
> Thank you so much for your valuable feedback and suggestions. Hope our rebuttal has addressed your concerns. We're glad to see that our feedback has resolved some of the concerns from Reviewer wBAQ and we are willing to actively address the concerns raised by any reviewers.
>
>
> We have revised our manuscript based on the current feedback. We have specified the changes in our revision in the previous General Response.  We hope to receive more constructive comments from all reviewers to further improve our paper.
>
>
> Thank you again for your time and efforts in assessing our paper.
>
>
> Best Regards, All Authors.

---

### Meta-Review · Area_Chair_xvaK · 2023-12-23

**Metareview:**

The paper introduces a novel diffusion-based method for generating synthetic training data, featuring three key innovations. It employs an MMD-based distribution matching objective for fine-tuning the diffusion model, incorporates visual guidance using intra-class mean features from CLIP, image captions, and class names to condition the model, and utilizes a VAE to obtain latent codes from real samples for initializing the latent prior of the diffusion model. Experimental evaluations confirm the effectiveness of the proposed approach across various settings and datasets.

The reviewers in this paper have a slight divergence of opinion, which was not changed by the rebuttal. However even Reviewer wBAQ with the most negative score acknowledged that there are interesting ideas in this paper and increased their score. After careful deliberation, we therefore decided to accept the paper. However, we strongly encourage the authors to improve the paper before the final version, taking into account especially the comments by Reviewer wBAQ and Reviewer BpUK.

**Justification For Why Not Higher Score:**

A borderline accept poster paper.

**Justification For Why Not Lower Score:**

N/A

---

### Decision · Program_Chairs · 2024-01-16

Accept (poster)